# Reversal of Pulmonary Fibrosis: Human Umbilical Mesenchymal Stem Cells from Wharton’s Jelly versus Human-Adipose-Derived Mesenchymal Stem Cells

**DOI:** 10.3390/ijms24086948

**Published:** 2023-04-08

**Authors:** Kuo-An Chu, Chang-Ching Yeh, Chun-Hsiang Hsu, Chien-Wei Hsu, Fu-Hsien Kuo, Pei-Jiun Tsai, Yu-Show Fu

**Affiliations:** 1Division of Chest Medicine, Department of Internal Medicine, Kaohsiung Veterans General Hospital, Kaohsiung 813414, Taiwan; kachu@vghks.gov.tw (K.-A.C.); chhsu1642@vghks.gov.tw (C.-H.H.); cwshe@vghks.gov.tw (C.-W.H.); 2School of Medicine, College of Medicine, National Sun Yat-sen University, Kaohsiung 804201, Taiwan; 3Department of Nursing, Shu-Zen Junior College of Medicine and Management, Kaohsiung 821004, Taiwan; 4School of Nursing, Fooyin University, Kaohsiung 831301, Taiwan; 5Department of Obstetrics and Gynecology, Taipei Veterans General Hospital, Taipei 112201, Taiwan; ccyeh39@gmail.com; 6Department of Obstetrics and Gynecology, National Yang Ming Chiao Tung University, Taipei 112304, Taiwan; 7Department of Nurse-Midwifery and Women Health, National Taipei University of Nursing and Health Sciences, Taipei 112303, Taiwan; 8Medical Intensive Unit, Department of Critical Care Medicine, Kaohsiung Veterans General Hospital, Kaohsiung 813414, Taiwan; 9Institute of Anatomy and Cell Biology, National Yang Ming Chiao Tung University, Taipei 112304, Taiwan; seraphy512@hotmail.com.tw; 10Trauma Center, Department of Surgery, Veterans General Hospital, Taipei 112201, Taiwan; 11Department of Critical Care Medicine, Veterans General Hospital, Taipei 112201, Taiwan; 12Department of Anatomy and Cell Biology, National Yang Ming Chiao Tung University, Taipei 112304, Taiwan

**Keywords:** pulmonary fibrosis, umbilical mesenchymal stem cells, Wharton’s jelly, adipose-derived mesenchymal stem cells, xenotransplantation

## Abstract

Pulmonary fibrosis (PF) is a progressive, non-reversible illness with various etiologies. Currently, effective treatments for fibrotic lungs are still lacking. Here, we compared the effectiveness of transplantation of human mesenchymal stem cells from umbilical cord Wharton’s jelly (HUMSCs) versus those from adipose tissue (ADMSCs) in reversing pulmonary fibrosis in rats. Bleomycin 5 mg was intratracheally injected to establish a severe, stable, single left lung animal model with PF. On Day 21 post-BLM administration, one single transplantation of 2.5 × 10^7^ HUMSCs or ADMSCs was performed. Lung function examination of Injury and Injury+ADMSCs rats displayed significantly decreased blood oxygen saturation and increased respiratory rates, while Injury+HUMSCs rats showed statistical amelioration in blood oxygen saturation and significant alleviation in respiratory rates. Reduced cell number in the bronchoalveolar lavage and lower myofibroblast activation appeared in the rats transplanted with either ADMSCs or HUMSCS than that in the Injury group. However, ADMSC transplantation stimulated more adipogenesis. Furthermore, matrix-metallopeptidase-9 over-expression for collagen degradation, and the elevation of Toll-like receptor-4 expression for alveolar regeneration were observed only in the Injury+HUMSCs. In comparison with the transplantation of ADMSCs, transplantation of HUMSCs exhibited a much more effective therapeutic effect on PF, with significantly better results in alveolar volume and lung function.

## 1. Introduction

Lung tissue destruction is caused by heterogeneous factors. The amount of functional alveoli is reduced due to lung impairment; consequently, fibrotic tissues gradually replace the alveolar compartments, resulting in pulmonary fibrosis (PF) development [1,2,3,4,5]. Given that pulmonary fibrosis is irreversible, PF patients suffer a progressive deterioration of lung functionality that causes respiratory failure as lung fibrosis aggravates, which may eventually lead to death. It was reported that patients have a median survival of three years once diagnosed with PF [6]. Currently, effective treatments for PF are still lacking. 

Investigations of mesenchymal stem cells for lung injury therapy are active in different areas [7,8,9,10]. Among them, adipose tissue mesenchymal stem cells (ADMSCs), abundant in adipose tissue, can be obtained with less invasive surgery. Results of case studies indicating that the transplantation of ADMSCs can alleviate inflammation, and thereby mitigate or prevent PF, have been reported [11,12,13,14,15,16]. Nevertheless, in most relevant studies, the therapeutic effects of ADMSCs were focused in acute pneumonia and preventing PF development by transplanting the cells either immediately, within one day, or during the acute phase of lung damage. As most PF patients seek treatment when they have already developed certain degrees of respiratory problems, with varying degrees of PF, how to reverse the pulmonary functionality in these patients is thus of critical importance.

Human umbilical mesenchymal stem cells (HUMSCs) are extracted from umbilical cord, which is generally considered to be waste after childbirth and can be obtained without invasive techniques. The number of HUMSCs collected can be expanded greatly in vitro. Moreover, xeno-transplanted HUMSCs are immunologically compatible with those of the receivers [17,18,19,20,21,22,23,24]. Therefore, HUMSCs can serve as an excellent stem cell resource in clinical medicine. In order to precisely evaluate the therapeutic effect of transplanted stem cells on chronic fibrosis stage or PF [25], and simultaneously to keep experimental animals alive, we established a severe, reproducible, consistent, one-sided left-lung PF animal model [17,18]. With this model, 2.5 × 10^7^ HUMSCs were intratracheally transplanted on Day 21 after BLM injection, and their therapeutic effects on pulmonary morphology and function were examined at one month post-transplantation [17,18].

The severe, reproducible, consistent, one-sided left-lung PF animal model was used in the present study. A single intra-tracheal administration of 2.5 × 10^7^ HUMSCs or ADMSCS was performed on Day 21 post-BLM treatment. The rats were examined for lung function once every week and sacrificed for investigation of lung morphology on Day 49 in order to compare the capability of HUMSCs and ADMSCs in the treatment of PF.

## 2. Results

### 2.1. Transplantation of HUMSCs, but Not ADMSCs, Improved Arterial Oxygen Saturation (SpO_2_) in Rats with PF

The SpO_2_ was measured via pulse oximetry to estimate the alveolar O_2_-CO_2_ exchange. The mean SpO_2_ stayed around 97.0 to 99.0% in the Normal group. The SpO_2_ of the Injury group significantly reduced to 81.6 ± 0.5% on Day 7 and persisted around 82.2 ± 0.8% until Day 49, which was markedly lower than that in the Normal group (Figure 1A,B, *p* < 0.05). The SpO_2_ levels in the Injury+ADMSCs and Injury+HUMSCs groups were comparable to those in the Injury group from Day 7 to Day 21. There was no increase in SpO2 for the Injury+ADMSCs group from Day 28 to Day 49. A significant increase in SpO_2_ to 85.3 ± 0.7% was found in the Injury+HUMSCs group on Day 28, and the trend persisted until it reached 92.3 ± 0.7% on Day 49, which was significantly better than those in the Injury and the Injury+ADMSCs groups (*p* < 0.05). Although the SpO_2_ in the Injury+HUMSCs group increased (from Day 28 to Day 49), it remained lower than the corresponding results in the Normal group (Figure 1A,B, *p* < 0.05).

### 2.2. Transplantation of HUMSCs, but Not ADMSCs, Attenuated the Rapid Respiratory Rate in Rats with PF

The lung function was also assessed by evaluating breaths per minute (BPM). The results showed that the respiratory rates kept steady, between 129 to 142 breaths/min, in the Normal group. In the Injury group, the respiratory rate significantly increased to 301.8 ± 7.6 breaths/min on Day 7 (*p* < 0.05) and remained high around 193.2 ± 5.8 breaths/min until Day 49 (*p* < 0.05). In the Injury+ADMSCs and the Injury+HUMSCs groups, the tendencies of rapid respiratory rate were comparable to those in the Injury group during Day 7 to Day 21. From Day 28 to Day 49, the respiratory rate was not ameliorated in the Injury+ADMSCs group. In the Injury+HUMSCs group, the respiratory rate significantly reduced since Day 28, in comparison with those in the Injury and the Injury+ADMSCs groups (*p* < 0.05). From Day 42 to Day 49, the difference in respiratory rate of the Injury+HUMSCs group and the Normal group was statistically insignificant (Figure 1C,D).

### 2.3. Transplantation of HUMSCs, but Not ADMSCs, Increased Body Weight in Rats with PF

Our findings showed that the Normal group’s body weight gradually increased over time. The body weight of the Injury group was stagnant around Day 7, and continued to increase after Day 7. In comparison with the Normal group, a significant reduction in weight appeared in the Injury group, and this trend persisted to Day 49 (*p* < 0.05). The trends of the body weight in the Injury+ADMSCs and the Injury+HUMSCs groups were comparable to those in the Injury group from Day 7 to Day 21. During Day 21 to Day 49, no significant difference was found in the Injury+ADMSCs group when comparing to the Injury group, and a considerable reduction was observed, in comparison with those in the Normal group. The Injury+HUMSCs group’s body weight substantially elevated from Day 28 to Day 49 when comparing with those in the Injury and the Injury+ADMSCs groups (*p* < 0.05) (Figure 2C).

### 2.4. Transplantation of HUMSCs, but Not ADMSCs, Enhanced Alveolar Space in Rats with PF

The changes in left-lung alveolar volume were calculated by MRI. With the existence of alveoli in both the left and right lungs, black signals were predominantly observed in the images of the Normal group (Figure 2A). Due to inflammation and immune cell infiltration, white signals were found in the left lung of the Injury group on Day 7. The alveolar space in the left lung was mostly occupied by consolidated tissues from Day 21 to Day 49 (Figure 2A,B). For the Injury+ADMSCs and the Injury+HUMSCs groups, white signals were found in the left lung from Day 7 to Day 21, suggesting the loss of alveolar volume and displacement by consolidated tissues. From Day 28 to Day 49, alveolar space in the left lung remained dominated by the white consolidated tissues in the Injury+ADMSCs group (Figure 2A,B). In the Injury+HUMSCs group, the alveolar space significantly elevated since Day 49, in comparison with those in the Injury and the Injury+ADMSCs groups (*p* < 0.05); however, statistical difference in alveolar volume between the Injury+HUMSCs group and the Normal group still existed (*p* < 0.05) (Figure 2A,B).

### 2.5. Transplantation of HUMSCs, but Not ADMSCs, Restored Alveolar Structure in Rats with PF

On Day 49 post-BLM administration, the left lung tissues were stained with H&E. In the Normal group, micrographs (with varying magnifications) showed an intact appearance of alveolar structure, with the connective tissues primarily surrounding the bronchus and rarely appearing among the alveoli. In the groups of Injury and Injury+ADMSCs, alveoli did appear in the peripheral region of the left lung, and a substantial amount of cells infiltrated in the central regions. However, areas with cell infiltration significantly reduced, and the alveolar space markedly increased in the central region of the Injury+HUMSCs group (*p* < 0.05) (Figure 3A–C). Subsequently, data from all left lung sections stained with H&E were summed up, and the quantitative results showed that left lung volume considerably decreased in the Injury group, as the volume of alveolar structure reduced, while 70% of the left lung’s volume was infiltrated with cells (*p* < 0.05) (Figure 3D–F). The left lung volume, the air space, and the cell infiltration area in the Injury+ADMSCs group were comparable to those in the Injury group (Figure 3D–F). In the Injury+HUMSCs group, the left lung volume and the air space were both close to those in the Normal group. In addition, the area of cell infiltration statistically decreased in the Injury+HUMSCs group, in comparison with those in the Injury and the Injury+ADMSCs groups (*p* < 0.05) (Figure 3D–F).

To evaluate the effectiveness of alveolar gas exchange, we also quantified the number and the circumference of alveoli per unit area in the outer region of the left lung where alveoli remained. Micrographs of the H&E-stained sections revealed that the alveolar size in the outer region of the left lung was smaller in the Normal group, implying that the number of alveoli per unit area was higher, and the total alveolar circumference for gas exchange was longer. A significantly larger alveolar size in the outer region in the Injury group than that in the Normal group was observed, which resulted in decreased total alveolar number and alveolar circumference per unit area (*p* < 0.05). The number of alveoli and the total alveolar circumference per unit area of the Injury+ADMSCs groups were not significantly different from those in the Injury group. Both parameters increased in the Injury+HUMSCs group in comparison with those of the Injury and the Injury+ADMSCs groups, suggesting that the transplantation of HUMSCs restored alveolar structure and recovered the efficiency of gas exchange (*p* < 0.05) (Figure 3C,G,H).

### 2.6. Transplantation of HUMSCs, but Not ADMSCs, Ameliorated Left Lung Shrinkage in Rats with PF

On Day 49, a global inspection of the both lungs revealed the presence of white alveolar structures with intact and smooth alveoli in the Normal group. On the contrary, the left lung markedly shrunk in the Injury group, with alveoli appearing only in the peripheral region and no alveoli but scar tissues found in the central region. In the Injury+ADMSCs group, the overall appearance of the left and right lungs showed no improvement. In the Injury+HUMSCs group, both the volume and the alveolar regions were notably preserved in the left lung by Day 49 (Figure 4A).

### 2.7. Transplantation of HUMSCs, but Not ADMSCs, Reduced Collagen Deposition in Rats with PF

The sections were stained with Sirius red to label collagen in the left lung (Figure 4B,C). Collagen predominantly appeared only in the regions surrounding the bronchus and blood vessels in the Normal group. In the Injury group, the red area of collagen significantly increased in the central region, which was comparable to the amount of collagen deposition in the Injury+ADMSCs group. The level of collagen deposition in the Injury+HUMSCs was lower than the corresponding level in the Injury and the Injury+ADMSCs groups (*p* < 0.05), and comparable to that in the Normal group (Figure 4B–D). We used Real-time RT-PCR to quantify the amount of rat *Col1a1* mRNA in fresh left lung on Day 49. The relative expression of rat *Col1a1* mRNA in the Injury and the Injury+ADMSCs groups increased significantly, in comparison with those in the Normal and the Injury+HUMSCs groups (*p* < 0.05) (Figure 4E).

### 2.8. Transplantation of HUMSCs, as Well as ADMSCs, Decreased Myofibroblast Amount in Rats with PF

Anti-α-SMA antibody was used to identify myofibroblasts (Figure 5A–C). In the Normal group, there were only scarce α-SMA-positive cells, primarily near the regions surrounding the bronchus. A significant amount of α-SMA-positive cells appeared in the connective tissues of the Injury group. There were only a few α-SMA-positive cells in the Injury+ADMSCs and the Injury+HUMSCs groups. The α-SMA expression level, assessed by Western blotting, showed that the sum of α-SMA was significantly higher in the Injury group in comparison with that in the Normal group (*p* < 0.05). In comparison with the Injury group, the α-SMA expression level was much lower in the Injury+ADMSCs and the Injury+HUMSCs groups (*p* < 0.05) (Figure 5A–C).

### 2.9. Transplantation of HUMSCs, as Well as ADMSCs, Diminished Cell Number in BALF in Rats with PF

The level of pulmonary inflammation, estimated via cell number measured in BALF, found that the cell count substantially increased in the Injury group on Day 49. The inflammatory cell count in the Injury+ADMSCs and the Injury+HUMSCs groups was lesser than that in the Injury group (*p* < 0.05), but comparable to that in the Normal group (Figure 5D). 

### 2.10. Transplantation of HUMSCs, but Not ADMSCs, Triggered the Activation of M2 Macrophages with Large Size in Rats with PF

Macrophage, identified via Anti-ED1 antibody, showed that only a few macrophages were present in the Normal group. In both the Injury and the Injury+ADMSCs groups, a significant number of smaller size macrophages were found in the connective tissues of the left lung. Larger size macrophages appeared in the connective tissues and among the alveolar space of left lung in the Injury+HUMSCs group (Figure 5E). Application of Anti-CD86 antibody to detect M1 macrophages found that the number of M1 macrophage was higher in the Injury group (*p* < 0.05). The number of M1 macrophage in Injury+ADMSCs group was lower than that in the Injury group, but higher than that in the Normal group (*p* < 0.05). The number of M1 macrophages significantly reduced in the Injury+HUMSCs group (Figure 5F,H). Application of Anti-CD163 antibody to detect M2 macrophages revealed that the number of M2 macrophages, which were rare in the Normal group, increased in the Injury and the Injury+ADMSCs groups. Additionally, the number of larger-size M2 macrophages was further amplified in the Injury+HUMSCs group (*p* < 0.05) (Figure 5G,I). 

### 2.11. Transplantation of HUMSCs, but Not ADMSCs, Stimulated MMP-9 Synthesis in Rats with PF

Application of Western blotting to assess the level of matrix metallopeptidase 9 (MMP-9) revealed no statistical differences among the Normal, the Injury, and the Injury+ADMSCs groups. The MMP-9 level distinctly increased only in the Injury+HUMSCs group (*p* < 0.05) (Figure 5J).

### 2.12. Transplantation of HUMSCs, but Not ADMSCs, Promoted TLR-4 Expression in Rats with PF

The protein level of TLR-4 was assessed by Western blotting. Among the Normal, the Injury, and the Injury+ADMSCs groups, the difference in the TLR-4 level was statistically insignificant. In contrast, the TLR-4 level in the Injury+HUMSCs group statistically increased (*p* < 0.05) (Figure 5K).

### 2.13. Transplantation of ADMSCs, but Not HUMSCs, Enhanced Adipogenesis in Rats with PF

Left lung sections acquired on Day 49 were stained with oil red O to label lipid droplets (Figure 6A–C). Micrographs (with varying magnifications) showed that only dye precipitation, rather than lipid droplets, appeared in the Normal group. In the Injury group, lipid droplets (stained in red) near the bronchus were found. In the Injury+ADMSCs group, the area of lipid droplets was more prominent and appeared everywhere. In the Injury+HUMSCs group, the amount of lipid droplets was less than that in the Injury+ADMSCs groups (*p* < 0.05) (Figure 6A–D).

## 3. Discussion

HUMSCs transplantation was more successful at reversing and repairing PF during the chronic phase of lung damage (i.e., in the fibrotic state) than ADMSCs transplantation was.

The majority of relevant studies focused on the therapeutic effects of ADMSCs or HUMSCs in acute pneumonia by transplanting the MSCs immediately, within one day, or during the acute phase of lung damage [11,12,13,14,15,16,26,27,28,29]. Their goals were to investigate the possibility of reducing lung inflammation, preventing the occurrence of acute respiratory distress syndrome (ARDS), decreasing the severity of ARDS, relieving the burden on the Intensive Care Unit, further reducing the mortality rate for patients with ARDS, and alleviating the development of pulmonary fibrosis. Most of the evidence has shown that transplantation of ADMSCs or HUMSCs during the acute phase of lung damage can lessen lung inflammation and lung fibrosis [11,12,13,14,15,16,26,27,28,29].

The purpose of this study was to compare the ability of HUMSCs and ADMSCs in the reversal or treatment of lung fibrosis (i.e., in the chronic phase of lung injury). At least three underlying therapeutic mechanisms of stem cells transplantation were involved in the reversal or treatment of PF [17]. First, stem cells inhibited inflammatory reactions and prevented further deterioration. Second, stem cells stimulated the host’s macrophages to produce MMP-9, which degraded the pre-existing collagen. Third, stem cells promoted the TLR-4 expression in the host’s alveolar epithelial cells and improved the HA-TLR-4 signaling for lung regeneration [17]. In this study, the αSMA, cell number in BALF, and M1 number in the rats’ left lung in the groups of Injury+ADMSCs and Injury+HUMSCs were significantly decreased in comparison with those in the Injury group on Day 49. Apparently, HUMSCs and ADMSCs appear to provide similar anti-inflammatory effects during the chronic phase of lung damage (i.e., in the fibrotic state).

Our previous studies showed that transplantation of HUMSCs stimulated the activation of autologous macrophages to synthesize MMP-9 and thereby assisted with the degradation of pre-existing collagen [17]. A similar conclusion was supported by Cabrera et al. [30]. Consistent with the results of Sirius red staining and *Col1a1* real-time RT-PCR in this study indicated that HUMSCs transplantation ameliorated collagen deposition. Here, we showed that ADMSCs implantation did not cause the host’s macrophages to produce MMP-9, which digested the pre-existing collagen. Hence, there were no statistical differences between the Injury and the Injury+ADMSCs groups in the results of Sirius red staining and *Col1a1* expression.

It is known that TLR-4 activation plays a critical part in regenerating and reconstituting alveolar epithelial cells [31]. Moreover, triggering of M2 polarization following the interactions of hyaluronan (HA)-TLR-4 was reported by Zhang et al. [32]. Consistently, we previously revealed that HUMSC transplantation robustly increased TLR-4 protein expression around the alveolar circumference and in M2 macrophages [17]. In this study, ADMSC transplantation did not promote TLR-4 expression in the left lung, and consequently, the regeneration of alveolar epithelial cells was not triggered. Similar conclusions have also been proposed by Uji et al. [33]. In their study, ADMSCs or saline were injected into rats two weeks after BLM injury. Their results showed that intravenous administration of ADMSCs did not ameliorate BLM-induced lung injury in rats [33]. Similar results obtained from clinical studies also supported that the transplantation of HUMSCs might indeed improve lung function in patients with idiopathic pulmonary fibrosis [34]. In the present study, ADMSC transplantation was administered 21 days following BLM damage, when collagen deposition had reached a plateau level. Our results showed that the difference in the levels of MMP-9 and TLR-4 between the Injury and the Injury+ADMSCs groups was statistically insignificant. The results suggested that ADMSC transplantation could neither restore nor reverse the lung function, lung fibrosis, and alveolar morphology when lung injury reached a stable and irreversible PF status. Several clinical trials have also confirmed that the transplantation of autologous ADMSCs did not significantly improve the lung function of idiopathic pulmonary fibrosis patients [35,36].

In the late stage or chronic stage of lung injury, transplantation of ADMSCs might cause another adverse complication, promoting the accumulation of a large number of fat cells throughout the fibrotic lung. We also observed fat accumulation in the left lung of the Injury group. It has been confirmed that the accumulation of visceral fat cells will augment the inflammation from organ damage [37,38,39]. A similar outcome, indicating that COVID-19 severity was associated with excess visceral adipose tissue, highlighted the potential pathogenic role of visceral adipose tissue in illness [40]. We suggest that ADMSCs may represent a double-edged sword in lung fibrosis, with anti-inflammatory effects on one hand and pro-inflammatory effects on the other. Likewise, it has been reported that ADMSCs may exhibit not only immunosuppressive properties [41,42], but also behave as an additional pathogenic source of pro-fibrosis in systemic sclerosis [43].

In addition to the above-mentioned factors, the observation that ADMSC transplantation might not improve lung fibrosis could be due to multiple factors, such as the timing of intervention. Previous studies showed that one injection of ADMSCs within two hours or two injections of ADMSCs within two hours and one week after irradiation resulted in more beneficial outcomes when compared to one late injection after seven days (ADMSC_7d_ group) or the Control (radiation alone) group [44]. We speculated that the difference in the pathological environment of the acute phase of lung injury vs. that of the chronic phase led to inconsistent therapeutic effects of ADMSCs. Another parameter that might influence the therapeutic potential of ADMSCs could be the age of the stem cell donors. Indeed, in one study comparing ADMSCs from aged (>22 months) vs. those from young (4 months) mice, only animals receiving young ADMSCs at day 21 after BLM instillation exhibited receded PF, less oxidative stress, and fewer markers of apoptosis compared to untreated controls [44]. In addition, MSCs isolated from various tissues exhibited similar morphology and surface marker expression, but even so they differed with regard to their differentiation potential and cytokine secretion [45]. As a result, MSCs from various origins may have different effects on organs. Mice were sensitized and challenged with ovalbumin to induce experimental allergic asthma. After twenty-four hours, mice received BMMSCs or ADMSCs intra-tracheally. Their results showed that animals receiving BMMSCs transplantation exhibited significantly greater reductions in lung inflammation compared to those in the groups of ADMSCs transplantation [46]. Similar to our findings, MSCs from different sources could affect lung injury or lung fibrosis differently.

## 4. Materials and Methods

This study uses the Materials and Methods from our previous studies, so the description in Materials and Methods partly reproduces the wording of the previous studies [17,18].

### 4.1. Establishment of Left-Lung PF Animal Model

After confirmation of anesthesia depth with Zoletil 20–40 mg/kg and Xylazine 5–10 mg/kg (intra-peritoneal injection), male SD rats, weighing approximately 250 g, received 5 Unit/5 mg BLM/250 g body weight (Nippon Kayaku Co., Ltd., Tokyo, Japan) in 200 μL phosphate-buffered saline by intratracheal injection, and were rotated to the left side by 60° for 90 min to establish a severe, reproducible, left-lung PF animal model [17,18].

### 4.2. Isolation and Culture of HUMSCs or Culture of ADMSCs

Isolation of HUMSCs in this study used the methods from our previous studies [17,18,19,20,21,22,23,24]. Human umbilical cords were collected and kept at 4 °C in Hank’s Balanced Salt Solution (HBSS). In a laminar hood, umbilical cords were disinfected by soaking in 75% ethanol, and then placed in the HBSS solution. Subsequently, the mesenchymal tissue (Wharton’s jelly) was cut into small pieces and centrifuged at 4000 rpm for 5 min. After removal of the supernatant fraction, the umbilical mesenchymal tissue was treated with collagenase and trypsin, followed by the addition of fetal bovine serum (FBS; Gibco 10437-028, Waltham, MA, USA) to stop the reaction. At this point, the umbilical mesenchymal cells were fully processed into human umbilical mesenchymal stem cells. Finally, HUMSCs were cultured in Dulbecco’s modified Eagle’s medium (DMEM) supplemented with 10% FBS or cryopreserved in liquid nitrogen for future use. The 15th passage HUMSCs were harvested for transplantation into rats.

Human adipose mesenchymal stem cells (ADMSCs) were purchased from Cellular Engineering Technologies (HMSC.AD-100, Coralville, IA, USA). ADMSCs were proliferated and sub-cultured in 10% FBS DMEM.

HUMSCs and ADMSCs were found to express high levels of CD44, CD73, CD90, and CD105 (Appendix A). Meanwhile, adipogenic and neuronal differentiation of HUMSCs and ADMSCs both succeeded (Appendix A).

### 4.3. Transplantation of ADMSCs and HUMSCs

Cultured ADMSCs and HUMSCs were treated with 0.05% Trypsin-EDTA (Gibco 15400-054, Dreieich, Germany) for 2.5 min. Cells were then collected, washed twice with 10% FBS DMEM, centrifuged at 1500 rpm for 5 min, and the supernatant was removed. The pelleted cells were then suspended in a concentration of 2.5 × 10^7^ in 200 μL of sterile saline solution. On Day 21 after intratracheal BLM, rats were treated with 2.5 × 10^7^ ADMSCs or HUMSCs by intratracheal transplantation.

### 4.4. Animal Groups

The animals were randomly selected for the following four treatment groups:Normal group (n = 15): Rats received saline injection. On Day 21 after saline injection, only 200 μL of saline was intratracheally administered (Figure 7A).Injury group (n = 15): Rats received 5 mg BLM injection. On Day 21 after BLM injection, no treatment but saline was intratracheally administered (Figure 7A).Injury+ADMSCs group (n = 15): Rats received BLM injection. Intratracheal transplantation of 2.5 × 10^7^ ADMSCs was carried out on Day 21 after BLM injection. The rats were sacrificed on Day 49 (Figure 7A).Injury+HUMSCs group (n = 15): Rats received BLM injection. Intratracheal transplantation of 2.5 × 10^7^ HUMSCs was carried out on Day 21 after BLM injection. The rats were sacrificed on Day 49 (Figure 7A).

Animals were randomized, blinded for treatments and assays. Parameters including body weight, blood oxygen saturation, respiratory rate, and MRI were evaluated once every week. On Day 49 post-BLM injury, rats were sacrificed for examination of lung morphology and bronchoalveolar lavage cell counts (Figure 7B).

### 4.5. Pulmonary Function Testing

#### 4.5.1. Arterial Blood Oxygen Saturation

The rats were deeply anesthetized by isoflurane (Baxter 228-194) after the fur overlaying their forelimbs was completely removed by shaving. A pulse oximeter (Pulseoximeter, NONIN LS1-10R) was then clipped onto the shaved forelimbs for quantifying arterial blood oxygen saturation (SpO_2_) [17,18].

#### 4.5.2. Pulmonary Respiratory Rates

The rats were put in a closed plethysmograph chamber (emka Technologies, Whole Body Plethysmograph) for 15 min to measure breathing volume and resting respiratory rate via the BIOPAC BSL 4.0 MP45 software [17,18].

### 4.6. Magnetic Resonance Imaging (MRI)

The MRI (BRUKER BIOSPEC 70/30) was used to obtain images of the rat lung at National Taiwan University’s Instrumentation Center. Horizontal scans of the rats’ thoracic cavity were performed at 1.5 mm intervals from rostral to caudal until the thoracic cavity was completely covered. As the first image collected in each rat differed in position, there were between 23 and 30 total MRI images in the horizontal plane.

The carina of trachea was used as a landmark for image positioning to reduce the bias caused by the different total number of scans when quantifying the left lung volume. Images of four slices below the carina were acquired in addition to the slice containing the carina. For the analysis of black alveolar space, data from these five images were summed, representing each rat’s left-lung alveolar volume.

### 4.7. Sacrifice and Perfusion Fixation of Rats

The rats were deeply anesthetized and transcardially perfused by 4% paraformaldehyde. Lung tissue blocks were sliced into serial sagittal sections in a thickness of 5 μm from the outermost lateral side. For subsequent histochemical or immunochemical staining, each set of tissue slices containing 10 consecutive slices, which were numbered and positioned on different glass slides (Appendix A).

### 4.8. Hematoxylin and Eosin (H&E) Staining

Lung tissue sections were mounted on gelatin-coated slides. After immersing the sections in hematoxylin solution (Muto Pure Chemicals Co., Ltd., Tokyo, Japan; No. 3008-1) and eosin solution (Muto Pure Chemicals Co., Ltd.; No. 3200-2), sections were dehydrated in ethanol, cleared in xylene (247642; Sigma-Aldrich, Darmstadt, Germany), and coverslipped with Permount (SP15-500; Thermo Fisher Scientific, Waltham, MA, USA). The left lung volume, percentage of cell infiltration area, and air space were quantified by the H&E-stained left lung sections [17,18].

### 4.9. Sirius Red Staining

Lung tissue sections were immersed with 0.1% Sirius red (Sigma-Aldrich, 2610-10-8) in picric acid. Sirius red-stained left lung sections were used to calculate the percentage of area occupied by collagen deposition [17,18].

### 4.10. Oil Red O Staining

Frozen left lung sections were stained with Oil Red O working solution (Sigma-Aldrich, O1391) for 10 min, and then counterstained with hematoxylin solution. Lipid droplets were stained red. The ratio of area occupied by lipid droplets in the left lung was quantified.

### 4.11. Immunohistostaining

Lung tissue sections were reacted with primary antibodies at 4 °C for 12 to 18 h, including mouse anti-α-smooth muscle actin antibody for identifying myofibroblast (SMA, Sigma-Aldrich, A2547); mouse anti-ED1 antibody for labeling total macrophage (Millipore, Burlington, MA, USA, MAB1435); rabbit anti-CD86 antibody for marking M1 macrophage (Proteintech, Rosemont, IL, USA, 13395-1-AP); and mouse anti-CD163 antibody for labeling M2 macrophage (BioRad, Hercules, CA, USA, MCA342R). Sections were then reacted with secondary antibodies, followed by avidin-biotinylated-HRP complex (ABC Kit, Vector Laboratories, Newark, CA, USA), and finally developed with 3,3′ -diaminobenzidine tetrahydrochloride hydrate (D5637, Sigma-Aldrich) as the chromogen [17,18].

### 4.12. Western Blotting

The protein expression was assessed by Western blotting. Primary antibodies, i.e., mouse anti-α-SMA antibody (SMA, Sigma-Aldrich, A2547), rabbit anti- MMP-9 antibody (Abcam, Cambridge, UK, ab76003), mouse anti-Toll-like receptor-4 (TLR-4) antibody (Abcam, ab30667), and mouse anti-β-actin antibody for internal control (Sigma-Aldrich, A5411) were reacted with the polyvinylidene fluoride (PVDF) membrane at 4 °C overnight. Subsequently, the corresponding secondary antibodies were immersed at room temperature for 1 h. The protein bands were quantified using Image J software and normalized using individual internal controls for comparison [17,18].

### 4.13. Bronchoalveolar Lavage Fluid (BALF)

The airway of each anesthetized rat was lavaged twice using 0.5 mL saline. Total cell number was counted by a hemocytometer [17,18].

### 4.14. Quantitative Real-Time PCR (qRT-PCR)

Total RNA was freshly isolated from the left lung by TRIzol reagent and reversely transcribed into complementary DNA with a PrimeScript RT reagent kit (BIONOVAS, North York, ON, Canada, HiScript I First Strand cDNA Synthesis Kit, AM0675-0050). SYBR-Green mix (Luna Universal qPCR Master Mix, M3003, New England Biolabs, Ipswich, MA, USA) was applied for quantitative PCR according to the manufacturer’s instructions. With β-actin levels as an internal control, the expression level of the target gene was normalized and calculated using the 2^−ΔΔCq^ method, and the relative mRNA expression was further calculated by normalizing to the Normal group.


Rat GAPDH
F: 5′-CTCTACCCACGGCAAGTTCAAC-3′
R: 5′-GGTGAAGACGCCAGTAGACTCCA-3′    Product length: 160 bpsRat collagen type1 alpha 1 chain (*Col1a1*)
F: 5′-TCCTGCCGATGTCGCTATC-3′
R: 5′-CAAGTTCCGGTGTGACTCGTG-3′    Product length: 234 bps


### 4.15. Statistical Analysis

All data are presented as the mean ± standard error of the mean (SEM). One-way or two-way analysis of variance was used to compare all means and Tukey’s multiple comparisons test for the posteriori test. A value of *p* < 0.05 was considered statistically significant.

## 5. Conclusions

In the late or chronic phase of lung injury, transplantation of ADMSCs did not improve lung function or reverse lung fibrosis. On the contrary, it enhanced the accumulation of fat cells in the lungs. In summary, factors including the implanted organ (site), the timing of intervention, the age of the donor, and the status of pathology need to be taken into account in the application of ADMSC transplantation. In comparison with ADMSC transplantation, transplantation of HUMSCs could be a better option to reverse and repair pulmonary fibrosis. Clinically, HUMSC transplantation may represent an effective modality for the treatment of patients with existing *PF* and COVID-19 sequela.

## Figures and Tables

**Figure 1 ijms-24-06948-f001:**
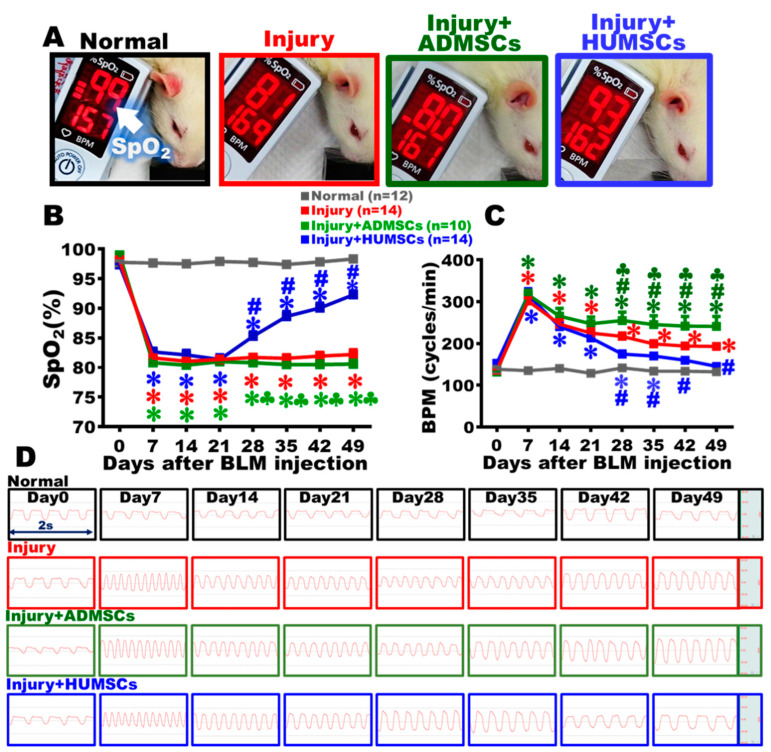
Transplantation of HUMSCs, but not ADMSCs, improved pulmonary function in rats with PF. On Day 49 after BLM damage, the rats in each group were examined for arterial SpO_2_ in their forelimbs (**A**). Weekly records show that the SpO_2_ level dramatically fell from Day 7 after BLM injury. Except for the Injury+HUMSCs group, the tendency of reduced SpO_2_ was maintained in the Injury and Injury+ADMSCs groups, up to Day 49 (**B**). The photographs show that each group’s respiratory frequency was recorded every 2 s from Day 0 to Day 49 (**D**). HUMSCs transplantation improved respiratory rate in rats with PF compared to the Injury and Injury+ADMSCs groups (**C**). The figure shows the number of rat samples in each group. ✽: *p* < 0.05 (vs. the Normal group); #: *p* < 0.05 (vs. the Injury group); ♣: *p* < 0.05 (the Injury+ADMSCs group vs. the Injury+HUMSCs group).

**Figure 2 ijms-24-06948-f002:**
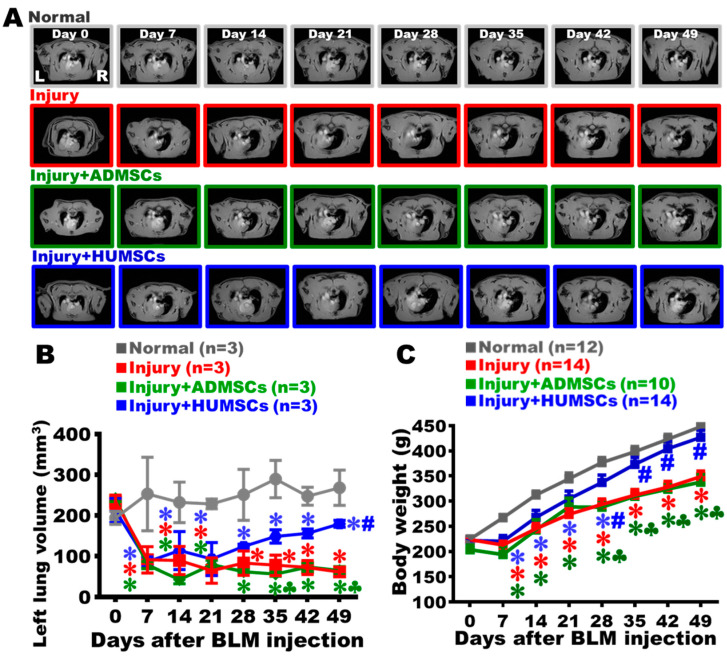
Transplantation of HUMSCs, but not ADMSCs, increased the alveolar volume in PF rats. The horizontal level of the trachea carina is shown in the MRI of the rat thoracic cavity. L stands for the left lung, R for the right lung (**A**). The carina was established as a reference point for image placement. In addition to the carina-containing slice, four images were acquired below the carina. To calculate the left-lung alveolar volume for each rat, the five pictures were added together and the black alveolar space was quantified. The black alveolar space in these five images was quantified to represent the left-lung alveolar volume of each rat. The findings indicated that the alveolar volume drastically decreased following BLM damage. The alveolar volume was significantly increased by the HUMSCs transplantation (**B**). Weekly weighing showed that HUMSCs transplantation improved the body weight of rats with PF (**C**). The figure shows the number of rat samples in each group. ✽: *p* < 0.05 (vs. the Normal group); #: *p* < 0.05 (vs. the Injury group); ♣: *p* < 0.05 (the Injury+ADMSCs group vs. the Injury+HUMSCs group).

**Figure 3 ijms-24-06948-f003:**
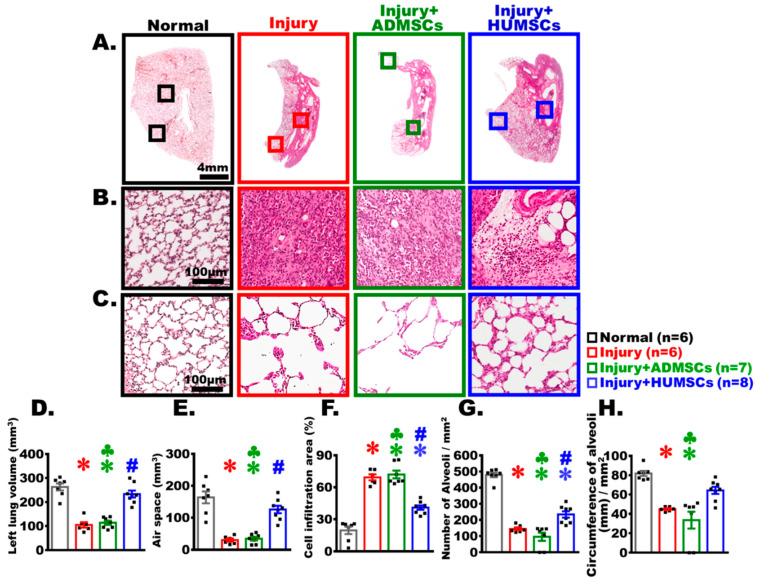
Transplantation of HUMSCs, but not ADMSCs, repaired alveolar structures in PF rats. Lower-magnification photographs of HE-stained left lung sections from each group were acquired on Day 49 (**A**). Photographs with higher magnification demonstrating a significant cell infiltration in the center regions of the left lungs following BLM injury. HUMSCs transplantation, but not ADMSCs transplantation, ameliorated cell infiltration in the central region (**B**). Photographs with higher magnification demonstrated that the alveolar structures were still present in the left lung’s periphery, despite the fact that the size of the alveoli varied (**C**). The total volume of the left lung, calculated by adding data from all HE-stained left lung sections, demonstrated that HUMSC transplantation significantly increased the total volume of the left lung (**D**) and the air space (**E**), while decreasing the cell inflammation areas (**F**). HUMSC transplantation effectively increased the number (**G**) and circumference (**H**) of alveoli per unit area for gas exchange, whereas the left lung structures of the Injury+ADMSCs group were similar to those of the Injury group. The figure shows the number of rat samples in each group. ✽: *p* < 0.05 (vs. the Normal group); #: *p* < 0.05 (vs. the Injury group); ♣: *p* < 0.05 (the Injury+ADMSCs group vs. the Injury+HUMSCs group).

**Figure 4 ijms-24-06948-f004:**
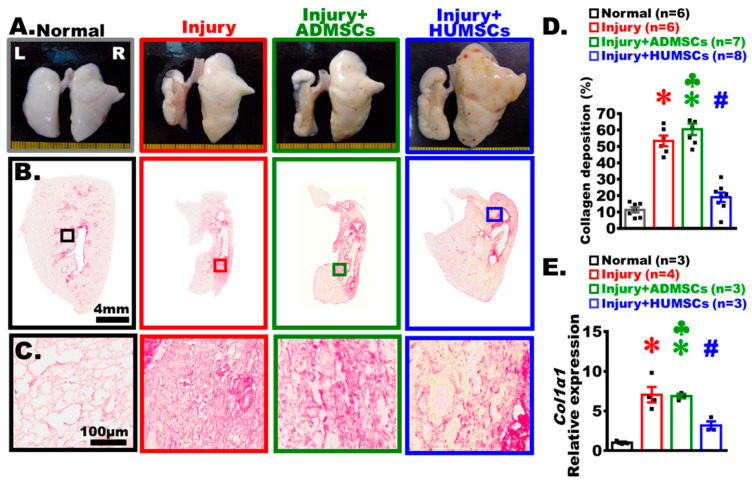
Transplantation of HUMSCs, but not ADMSCs, reduced collagen accumulation in PF rats. Photographs of the lungs’ anterior views were acquired from each group on Day 49. The left lungs clearly shrank, and alveoli were only seen in the peripheral region of the left lung, with the exception of the Injury+HUMSCs group (**A**). Lower-magnification (**B**) and higher-magnification (**C**) photographs of Sirius-red-stained left lung sections from each group were acquired on Day 49 to highlight the presence of collagen. Quantification of the percentage of area occupied by collagen in the rats’ left lung revealed that HUMSCs transplantation alleviated the fibrosis in rats with PF, in comparison with those in the Injury and the Injury+ADMSCs groups (**D**). On Day 49, rat *Col1a1* mRNA was collected from the left lung and examined with qRT-PCR. The *Col1a1* expression in the Injury+HUMSCs group was significantly lower than those in the Injury and Injury+ADMSCs groups. The figure shows the number of rat samples in each group (**D**,**E**). ✽: *p* < 0.05 (vs. the Normal group); #: *p* < 0.05 (vs. the Injury group); ♣: *p* < 0.05 (the Injury+ADMSCs group vs. the Injury+HUMSCs group).

**Figure 5 ijms-24-06948-f005:**
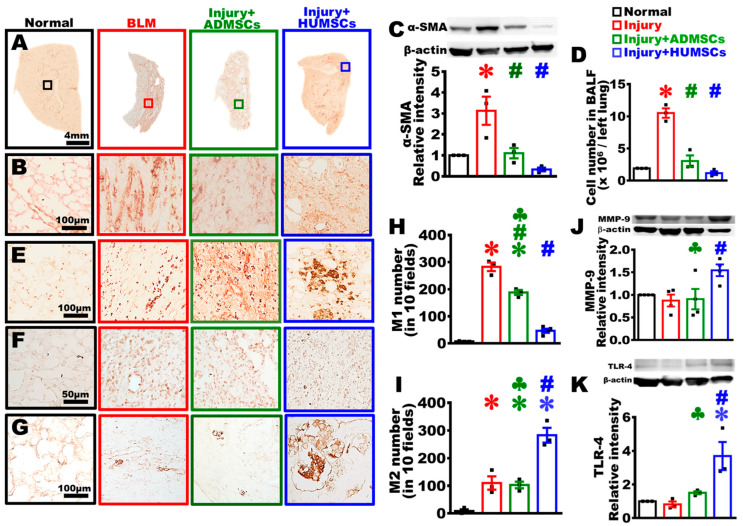
Transplantation of HUMSCs, but not ADMSCs, stimulated MMP-9 synthesis and TLR-4 expression in PF rats. Lower-magnification (**A**) and higher-magnification (**B**) immunohistochemical photographs of left lung sections from each group were stained with anti-α-SMA antibody to detect activated myofibroblasts on Day 49. The amount of α-SMA in the left lung was measured using a Western blot, and it revealed that the Injury+ADMSCs and Injury+HUMSCs groups had much less activated myofibroblasts than the Injury group did. On Day 49 (**C**). The cell count of bronchoalveolar lavage remained greater in the Injury group than those in the Injury+ADMSCs and Injury+HUMSCs groups (**D**). On Day 49, immunohistochemical photographs of left lung sections from each group were stained with anti-ED1 (**E**), anti-CD86 (**F**), and anti-CD163 antibodies (**G**) to label macrophage, M1 macrophage, and M2 macrophage, respectively. Quantification results showed that an increase in M1 macrophages in the Injury and the Injury+ADMSCs groups (**H**). Moreover, a robust increase in M2 macrophages in the Injury+HUMSCs group (**I**). Measurement of the amount of MMP-9 (**J**) and TLR-4 (**K**) expression in the left lung using Western blotting revealed that HUMSCs transplantation increased MMP-9 and TLR-4 expression. n = 6~8 in each group for immunostaining, n = 3~4 in each group for Western blotting. ✽: *p* < 0.05 (vs. the Normal group); #: *p* < 0.05 (vs. the Injury group); ♣: *p* < 0.05 (the Injury+ADMSCs group vs. the Injury+HUMSCs group).

**Figure 6 ijms-24-06948-f006:**
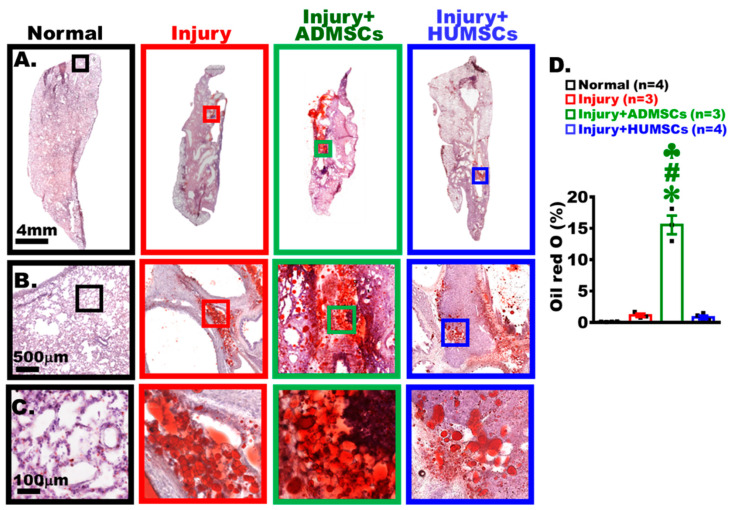
Transplantation of ADMSCs, but not HUMSCs, enhanced adipocyte accumulation in rats with PF. Photographs (with various magnifications) of left lung sections from the Normal, the Injury, the Injury+ADMSCs, and the Injury+HUMSCs groups on Day 49, stained with oil red O to label adipocytes (**A**–**C**), showed a large number of adipocytes in the left lung of the Injury+ADMSCs group (**D**). ✽: *p* < 0.05 (vs. the Normal group); #: *p* < 0.05 (vs. the Injury group); ♣: *p* < 0.05 (the Injury+ADMSCs group vs. the Injury+HUMSCs group).

**Figure 7 ijms-24-06948-f007:**
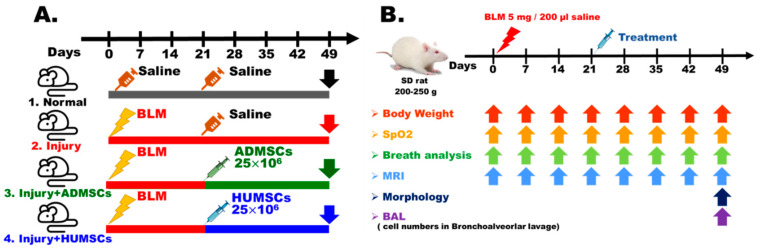
Scheme of experimental procedures and animal grouping. The rats were divided into four groups (**A**). Day 0 was defined as the day that BLM was administrated to the rats. HUMSCs or ADMSCs were delivered on Day 21 after BLM injection. Rats were sacrificed on Day 49 post-BLM injury (**B**).

## Data Availability

The corresponding author, YSFu, will make all the data behind the study’s conclusions available to the public under reasonable request.

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
