# Peer review of "Reversal of Pulmonary Fibrosis: Human Umbilical Mesenchymal Stem Cells from Wharton’s Jelly versus Human-Adipose-Derived Mesenchymal Stem Cells"

_ijms, 2023, doi:10.3390/ijms24086948_

Round 1

Reviewer 1 Report

Comments to the Author

The authors showed that HUMSCs can improve lung inflammation and fibrosis in bleomycin-induced lung injury model. This study has novelty and clinical significance. The following issue should be resolved for the acceptance of IJMS.

As the authors say, the timing of interventional drug administration is important in the bleomycin model. Therefore, I believe that the efficacy of interventional drug should be compared among groups treated with interventional drug at multiple points. It would be better if efficacy of HUMSCs were clarified in the acute phase of the bleomycin model (within 7 days after administration of bleomycin) is compared with that of groups treated at 21 day after administration of bleomycin. If not, please show previous data for the acute intervention group or, if not possible, the need for the above studies and limitation of the present study in discussion section.

Author Response

Comments to the Author

The authors showed that HUMSCs can improve lung inflammation and fibrosis in bleomycin-induced lung injury model. This study has novelty and clinical significance. The following issue should be resolved for the acceptance of IJMS.

As the authors say, the timing of interventional drug administration is important in the bleomycin model. Therefore, I believe that the efficacy of interventional drug should be compared among groups treated with interventional drug at multiple points. It would be better if efficacy of HUMSCs were clarified in the acute phase of the bleomycin model (within 7 days after administration of bleomycin) is compared with that of groups treated at 21 day after administration of bleomycin. If not, please show previous data for the acute intervention group or, if not possible, the need for the above studies and limitation of the present study in discussion section.

ANS 1:    

As suggested, we have added three paragraphs in the Discussion (from page 17, Lines 383~ 436).

ANS 2:   

As suggested, four references about HUMSCs transplantation in the acute phase were added (as references No 26- 29).

Reviewer 2 Report

In a paper by K-A Chu et al. the efficacy of placental vs adipose-derived MSCs were compared in a rat model of bleomycin pulmonary fibrosis. The work is relevant and interesting, despite the fact that such studies have been carried out over the past 10 years. At the same time, there are a number of comments that should be addressed before the article is accepted for publication. For instance, the result shown by the authors, with predominant adipogenic differentiation of ADMSCs after endothracheal implantation, looks quite convincing; however, other histological data require supplementation.

1.       Line 64-65. Authors say: «Currently, effective treatments for PF are still lacking». That is not correct since Pirfenidone and Nintedanib has been approved by FDA for IPF treatment.

2.       In the Introduction talking about the use of MSCs for the treatment of idiopathic pulmonary fibrosis, the authors refer several preclinical studies, without mentioning the published results of clinical trials, that is not correct (https://pubmed.ncbi.nlm.nih.gov/31613055/, https://pubmed.ncbi.nlm.nih.gov/34195252/).

3.       An experimental treatment of bleomycin pulmonary fibrosis in rats in 2023 without a positive control group treated with pirfenidone looks strange. What if the efficacy of pirfenidone is much higher than that of the studied HUMSCs? I would firmly suggest using an additional control group with pirfenidone treatment.

4.       It is very strange that ADMSCs did not show any regenerative effect. The question arises, whether the adipose derived preparations contain MSCs at all? What is the amount of CD29+, CD73+, CD90+, and CD105+ cells in these preparations? Was it the same with HUMSCs? Were MSCs from adipose tissue initially multipotent?

5.       Fig. 3A as well as Suppl. Fig 1-4 are not informative at all.

6.       Panels C on Fig. 4 do not differ from each other in groups HUMSCs and ADMSCs. Panels B are not representative. The low magnification images show that central displacement of panel B in the HUMSCs group will show the same interstitial infiltration as in the ADMSCs group. More convincing histological data should be presented (several fields of view from several animals).

7.       According macro view in Fig.5 the fibrosis was the same in control group as well as in ADMSCs and HUMSCs groups. More convincing histological data should be presented.

Author Response

Comments and Suggestions for Authors

In a paper by K-A Chu et al. the efficacy of placental vs adipose-derived MSCs were compared in a rat model of bleomycin pulmonary fibrosis. The work is relevant and interesting, despite the fact that such studies have been carried out over the past 10 years. At the same time, there are a number of comments that should be addressed before the article is accepted for publication. For instance, the result shown by the authors, with predominant adipogenic differentiation of ADMSCs after endothracheal implantation, looks quite convincing; however, other histological data require supplementation.

  1. Line 64-65. Authors say: «Currently, effective treatments for PF are still lacking». That is not correct since Pirfenidone and Nintedanib has been approved by FDA for IPF treatment.

ANS 1: 

Nintedanib and Pirfenidone are approved medications known to decrease pulmonary fibrosis progression.

ANS 2: 

At least, three underlying therapeutic mechanisms of stem cells transplantation may involve in the reversal or treatment of pulmonary fibrosis. First, stem cells inhibited inflammatory reactions and prevented further deterioration. Second, stem cells stimulated the host’s macrophages to produce MMP-9, which degraded the pre-existing collagen. Third, stem cells promoted the TLR-4 expression in the host’s alveolar epithelial cells and improved the HA-TLR-4 signaling for lung regeneration.

ANS 3: 

Our previous study showed that the cell number in bronchoalveolar lavage in the BLM+Nintedanib or BLM+Pirfenidone groups was significantly lower than that of the BLM group, indicating Nintedanib or Pirfenidone reduced inflammatory responses in the chronic stage.

ANS 4: 

Western blotting was applied to quantify the expression of matrix metallopeptidase 9 (MMP-9). Our previous results showed that no statistical differences were identified among the BLM, BLM+Nintedanib, and BLM+Pirfenidone groups. The MMP-9 level markedly elevated in the BLM+HUMSCs group, suggesting transplantation of HUMSCs increased MMP-9 expression, which could help collagen degradation in the fibrotic region of lung

ANS 5: 

The protein level of toll-like receptor-4 (TLR-4) was quantified using Western blotting. Our previous results showed that no statistical significance in TLR-4 expression was observed between the Normal, BLM, BLM+Nintedanib, and BLM+Pirfenidone groups.

ANS 6: 

In summary, Nintedanib or Pirfenidone could not restore or reverse the damaged lung tissues when pulmonary damage reached a stable and irreversible PF status. Nevertheless, Nintedanib or Pirfenidone reduced inflammatory responses in the chronic stage.

  1. In the Introduction talking about the use of MSCs for the treatment of idiopathic pulmonary fibrosis, the authors refer several preclinical studies, without mentioning the published results of clinical trials, that is not correct (https://pubmed.ncbi.nlm.nih.gov/31613055/, https://pubmed.ncbi.nlm.nih.gov/34195252/).

ANS 1: 

As suggested, these two references have been cited in the Introduction (page 3, Line 67) and added in References as reference No 9 and reference No 10 (page 30, Lines 698~ 705).

  1. An experimental treatment of bleomycin pulmonary fibrosis in rats in 2023 without a positive control group treated with pirfenidone looks strange. What if the efficacy of pirfenidone is much higher than that of the studied HUMSCs? I would firmly suggest using an additional control group with pirfenidone treatment.

ANS 1: 

We have completed the experiments which is in line with the reviewer’s suggestion. For our related experiments and published paper, please refer to the following article.

Comparison of reversal of rat pulmonary fibrosis of nintedanib, pirfenidone, and human umbilical mesenchymal stem cells from Wharton's jelly. Stem Cell Res Ther. 2020; 11(1): 513. doi: 10.1186/s13287-020-02012-y. 

ANS 2: 

The above paper was cited as reference No 18 in this study.

  1. It is very strange that ADMSCs did not show any regenerative effect. The question arises, whether the adipose derived preparations contain MSCs at all? What is the amount of CD29+, CD73+, CD90+, and CD105+ cells in these preparations? Was it the same with HUMSCs? Were MSCs from adipose tissue initially multipotent?

ANS 1: 

At least, three underlying therapeutic mechanisms of stem cells transplantation may involve in the reversal of PF. First, stem cells inhibited inflammatory reactions and prevented further deterioration. Second, stem cells stimulated the host’s macrophages to produce MMP-9, which degraded the pre-existing collagen. Third, stem cells promoted the TLR-4 expression in the host’s alveolar epithelial cells and improved the HA-TLR-4 signaling for lung regeneration.

ANS 2: 

In this study, ADMSCs have only anti-inflammatory effects in fibrotic lungs, do not degrade fibrotic tissue, and do not promote the regeneration of alveolar epithelial cells. As a result, they may only inhibit lung deterioration and may not repair the fibrotic lungs.

ANS 3: 

The related descriptions have added in the Discussion (from page 18, Lines 394~ 436)

ANS 4: 

As suggested, the descriptions about surface marker analysis and multipotent differentiation of ADMSCs and HUMSCs have been added in the Materials and Methods (page 22, Lines 501~ 503) and Supplemental Figure 1 (page 37, Lines 843~ 851).

  1. Fig. 3A as well as Suppl. Fig 1-4 are not informative at all.

ANS 1: 

As suggested, we have deleted the original Supplemental Figure 1- 4 in this revised manuscript.

ANS 2: 

The MRI images of the rat's horizontal thoracic cavity at the level of trachea carina from each group were acquired as representatives on Day 0, 7, 14, 21, 28, 35, 42, and Day 49, respectively. Due to inflammation and immune cell infiltration, white signals (consolidated tissues) were found in the left lung of the Injury and Injury+ADMSCs groups from Day 21 to Day 49. However, the alveolar space (black signals) still were found in the left lung of the Injury+HUMSCs group from Day 21 to Day 49 (Figure 3A)

ANS 3: 

The related descriptions for Figure 3A were remodeled and showed in the Results (page7, Lines 155~ 170).

  1. Panels C on Fig. 4 do not differ from each other in groups HUMSCs and ADMSCs. Panels B are not representative. The low magnification images show that central displacement of panel B in the HUMSCs group will show the same interstitial infiltration as in the ADMSCs group. More convincing histological data should be presented (several fields of view from several animals).

ANS 1: 

Panels C in Fig. 4, In order to evaluate the effectiveness of alveolar gas exchange, we also quantified the number and the circumference of alveoli per unit area in the outer region of left lung where alveoli remained. Micrographs of the H&E stained sections revealed that the alveolar size in the outer region of left lung was smaller in the Normal group, implying that the number of alveoli per unit area was higher, and the total alveolar circumference for gas exchange was longer. A significantly larger alveolar size in the outer region in the Injury and Injury+ADMSCs groups than that in the Normal group was observed, which resulted in decreased total alveolar number and alveolar circumference per unit area (p<0.05). Both parameters increased in the Injury+HUMSCs group in comparison with those of the Injury and the Injury+ADMSCs groups, suggesting that transplantation of HUMSCs restored alveolar structure and recovered the efficiency of gas exchange (p<0.05) (Figures 4C, 4G- 4H).

ANS 2: 

The related descriptions for Figure 4C were showed in the Results (page 11, Lines 229~242).

ANS 3: 

We have changed photographs in Figures 4B and 4C of the Injury+HUMSCs group with clearer images (page 10, Line 208).

  1. According macro view in Fig.5 the fibrosis was the same in control group as well as in ADMSCs and HUMSCs groups. More convincing histological data should be presented.

ANS 1: 

We have changed photographs in Figures 5B and 5C of the Injury+ADMSCs Injury+HUMSCs groups with clearer images (page 12, Line 254).

Round 2

Reviewer 1 Report

Author correctly revised the manuscript.

Reviewer 2 Report

The authors have addressed all my comments and answered to my questions. The revised paper can be accepted.